# Evaluation of left ventricular ejection fraction by a new automatic tool on a pocket ultrasound device: Concordance study with cardiac magnetic resonance imaging

**Lucie Berger[1], Fabien Coisy[1], Skander Sammoud[2], Fabien de Oliveira[2], Romain Genre Grandpierre[1], Laura Grau-Mercier[1], Xavier Bobbia[3], Thibaut Markarian[4]\***

**1** Department of Emergency Medicine, UR UM 103 (IMAGINE), Nîmes University Hospital, Montpellier University, Nîmes, France, **2** Department of Medical Imaging, IPI Platform, Nîmes University Hospital, Medical Imaging Group Nîmes, IMAGINE, University of Montpellier, Nîmes, France, **3** Department of Emergency Medicine, UR UM 103 (IMAGINE), Montpellier University Hospital, Montpellier University, Montpellier, France, **4** Department of Emergency Medicine, UMR 1263 (C2VN), Assistance Publique des Hôpitaux de Marseille (APHM), Timone University Hospital, Aix-Marseille University, Marseille, France

\* thibaut.markarian@ap-hm.fr

## Abstract

### Introduction

Assessment of left ventricular ejection fraction (LVEF) is one of the primary objectives of echocardiography. The gold standard assessment technique in emergency medicine is eye-balling. A new tool is now available on pocket ultrasound devices (PUD): automatic LVEF. The primary aim of this study was to evaluate the concordance between LVEF values estimated by automatic LVEF with PUD and by cardiac magnetic resonance imaging (MRI).

### Materials

This was a prospective, monocentric, and observational study. All adult patients with an indication for cardiac MRI underwent a point-of-care ultrasound. Blinded to the MRI results, the emergency physician assessed LVEF using the automatic PUD tool and by visual evaluation.

### Results

Sixty patients were included and analyzed. Visual estimation of LVEF was feasible for all patients and automatic evaluation for 52 (87%) patients. Lin's concordance correlation coefficient between automatic ejection fraction with PUD and by cardiac MRI was 0.23 (95% CI, 0.03–0.40).

### Conclusion

Concordance between LVEF estimated by the automatic ejection fraction with PUD and LVEF estimated by MRI was non-existent.

**Data Availability Statement:** All relevant data are within the paper and its Supporting Information files.

**Funding:** The author(s) received no specific funding for this work.

**Competing interests:** I have read the journal's policy and the authors of this manuscript have the following competing interests: TM and XB declare a competing interest as US teachers for GE (GE MEDICAL SYSTEMS ULTRASOUND) customers. The other authors state they have no competing interests. This does not alter our adherence to PLOS ONE policies on sharing data and materials.

## Introduction

Point-of-care ultrasound (POCUS) is now a tool used daily in emergency medicine for patient management [1, 2] and echocardiography is one of the most widely used [2, 3]. Assessment of left ventricular ejection fraction (LVEF) is one of the primary objectives of echocardiography. The gold standard assessment technique in emergency medicine is eyeballing [1, 4, 5]. This technique requires minimal training in image acquisition and interpretation [6]. Its perfectible relevance and reproducibility has led to the use of other ultrasound tools in acute patients [7]. These correspond to the measurements of the Mitral Annular Plan Systolic Excursion (MAPSE) [8] or the peak systolic mitral annular velocity (Sa) calculated by tissue Doppler imaging [9]. These techniques require training, a suitable ultrasound system, and time to perform the measurements.

In the last few years, pocket ultrasound devices (PUD) have been developed [10]. They have shown their relevance for focused cardiac ultrasonography in an emergency setting [11] and even in adverse conditions [12]. Thus, the academic societies of cardiology have published recommendations on their use [13]. Automatic LVEF assessment is now available on PUD. Based on a simplified Speckle tracking technique, the device allows the calculation of LVEF based on a difference in left intraventricular volume area from a single cardiac view. The relevance of this tool is an essential prerequisite for its use.

Our hypothesis is that the agreement between LVEF estimated automatically by PUD and LVEF estimated by cardiac magnetic resonance imaging (MRI) is good. The primary aim of this study was to evaluate concordance between LVEF values estimated by automatic LVEF (Auto-EF) with PUD and those obtained by cardiac MRI.

## Materials and methods

### Study design

This was a prospective, monocentric, observational study conducted in the emergency department (ED) of the University Hospital of Nîmes. This study was carried out in accordance with the amended Declaration of Helsinki. In accordance with French law [14], the study protocol was approved by the *Comité de protection des personnes* (CPP Nord-Ouest IV, ref 20.05.20.57844). Written informed consent was obtained from all participants and/or their legal guardians. Recruitment for this study began on June 06, 2021 and ended on July 06, 2022.

### Patients

All adult patients undergoing cardiac MRI were eligible for enrollment only when a study emergency physician was available. Because LVEF measurement on MRI was systematic, patients were included regardless of the indication for cardiac MRI. Patients could be enrolled if they were at least 18 years of age, were not pregnant or under legal guardianship, and if informed consent was obtained.

### Data collection and measurements

Epidemiological data collected included gender, age, medical history of cardiovascular problems, vital signs on hospital admission, and the indication for cardiac MRI. Ultrasound data and bedside interpretations were recorded at the time of performing cardiac POCUS.

## Ultrasound techniques

POCUS was performed in the first hour after the cardiac MRI by emergency physicians (EPs) who were informed about the protocol but blinded to the MRI results. They all had university training in POCUS and usually used all the protocol measurements in their clinical practices. US scans were performed with a PUD: V-Scan Dual Probe (GE Medical System, Milwaukee, Wisconsin, USA). The LVEF measurements were performed with a phased array probe and using the apical four-cavities view. The software automatically identifies the left ventricular (LV) endocardial border, measures end-systolic and end-diastolic volumes, and automatically calculates LVEF (Auto-EF) [15]. The classification of heart failure according to the ejection fraction (EF) selected for the concordance analysis were preserved (LVEF $\geq$ 50%), mid-range (LVEF 40–49%), and reduced (LVEF < 40%) [4].

After performing the POCUS with PUD, a second scan was performed with a conventional US device: Venue (GE Medical Systems, Milwaukee, Wisconsin, USA) and a phased array probe. MAPSE was measured in a time-motion mode at the lateral mitral ring [16]. The classification of EF according to MAPSE considered for the concordance analysis were MAPSE < 7 mm (reduced LVEF), MAPSE between 7 and 10 mm (mid-range LVEF), and MAPSE > 10 mm (preserved LVEF) [16]. Tissue Doppler imaging with the sample volume was placed in the lateral mitral ring to record the peak systolic mitral annular velocity: Sa [16]. The classification of EF according to Sa used for the concordance analysis were Sa < 6 cm.s$^{-1}$ (reduced LVEF), Sa between 6 and 8 cm.s$^{-1}$ (mid-range LVEF), and Sa $\geq$ 9 cm.s$^{-1}$ (preserved LVEF) [17].

The quality of images obtained by two ultrasound devices was recorded according to this classification: 1: no image; 2: insufficient image quality for visual interpretation; 3: poor image quality; 4: adequate image quality; 5: excellent image quality [18].

No adjustment was made for patients in atrial fibrillation. However, Sa measurement corresponded to the mean of 3 measurements in the presence of atrial fibrillation.

## Cardiac MRI technique

Scans were performed using an IRM Siemens magnetom sola 1.5 T MRI scanner. Retrospective gating was triggered by the ECG in cases of sinus rhythm, and prospective gating in cases of significant arrhythmia. Volumetric coverage of the LV using short-axis cines and the disc summation method were used to measure LVEF. To obtain a smooth contour of the endocardial borders, the LV papillary muscles were included in the LV end-diastolic and systolic volumes.

## Aims and outcomes

The main objective of this study was to evaluate the concordance between LVEF values estimated by automatic LVEF (Auto-EF) with PUD and by cardiac MRI.

Secondary objectives were: (1) to assess the concordance between three heart failure classes according to EF determined by Auto-EF and MRI; (2) to assess the concordance between and the LVEF determined by visual estimating with PUD and MRI; (3) to evaluate the concordance between three heart failure classes according to EF determined by visual estimating LVEF with PUD and MRI; (4) to evaluate the correlation between the mitral annular plane systolic excursion (MAPSE) value measured with a conventional ultrasound (US) device and the LVEF estimated by MRI; (5) to evaluate the concordance between three heart failure classes according to EF determined by MAPSE and MRI; (6) to evaluate the correlation between the peak systolic mitral annular velocity (Sa) measured with a conventional US device and the LVEF estimated by MRI; and (7) to evaluate the concordance between three heart failure classes according to EF determined by Sa and MRI.

### Number of subjects

To determine the number of subjects required, we performed a retrospective analysis of 75 patients who underwent cardiac MRI at our center: the mean LVEF was 50% ± 15%. Because we had no reason to think that the PUD would make a systematic error, we hypothesized that the auto-EF would average 50% ± 15%. Our hypothesis was that the concordance between auto-EF and LVEF estimated by MRI was good (Lin's coefficient ≥ 70%) [15]. Furthermore, we thought that the correlation between the two measures would be strong (Pearson coefficient > 0.8). To estimate the expected value of a Pearson coefficient at 0.85 and Lin's coefficient at 0.80 with a lower bound of the confidence interval at 0.70, we needed 52 patients. Assuming a rate of 15% lost to follow up, 60 patients needed to be included.

### Data analysis

The quantitative variables are expressed as means and standard deviations or as medians and 25th and 75th percentiles, depending on their distribution. For the qualitative variables, counts and associated percentages are presented. The quantitative variables across the groups were compared using the Student's (or Kruskal–Wallis) test. The relationship between the qualitative variables was tested using a chi-squared test (or Fisher's exact test). Concordance of two LVEF measurements was assessed using Lin's concordance coefficient along with its 95% CI. The linear correlation of two LVEF measurements or of a LVEF measurement and another evaluation tool (MAPSE and Sa) was analyzed by the Pearson correlation coefficient. The evaluation of concordance was completed by the search for a possible bias by the graphical method of Bland and Altman. Agreement between the three heart failure classes according to EF was assessed by Cohen's kappa coefficient. The significance level was set at 5%. Statistical analyses were performed with SAS software version 7.1 (SAS Institute, Cary, NC, USA).

## Results

### Participants

From June 2021 to July 2022, 60 patients were included and analyzed (Fig 1). Patients' characteristics at inclusion are shown in Table 1.

The quality of images obtained with PUD was adequate for 23 (38%) patients, poor for 19 (32%), excellent for 16 (27%), and insufficient for visual interpretation for 2 (3%). Visual estimation of LVEF was feasible for all patients and automatic evaluation for 52 (87%) patients.

### Primary outcome

Lin's concordance correlation coefficient between automatic EF with PUD and by cardiac MRI was 0.23 (95% CI, 0.03–0.40), whereas the Pearson correlation coefficient was 0.33 (95% CI, 0.06–0.56) (Fig 2).

### Secondary outcomes

Concordance between auto-EF with PUD and by cardiac MRI for classification into three stages of heart failure was 0.16 (95% CI, 0.00–0.33) (Table 2). Lin's concordance correlation coefficient between visual estimating LVEF with PUD and the LVEF estimated by MRI was 0.68 (95% CI, 0.51–0.79), whereas the Pearson correlation coefficient was 0.69 (95% CI, 0.52–0.80) (Fig 2). Concordance between visual estimating LVEF with PUD and the LVEF estimated by cardiac MRI for classification into three stages of heart failure was 0.55 (95% CI, 0.35–0.75) (Table 2). The Pearson correlation coefficient between MAPSE and the LVEF estimated by cardiac MRI was 0.55 (95% CI, 0.25–0.75) (Fig 3). Concordance between MAPSE

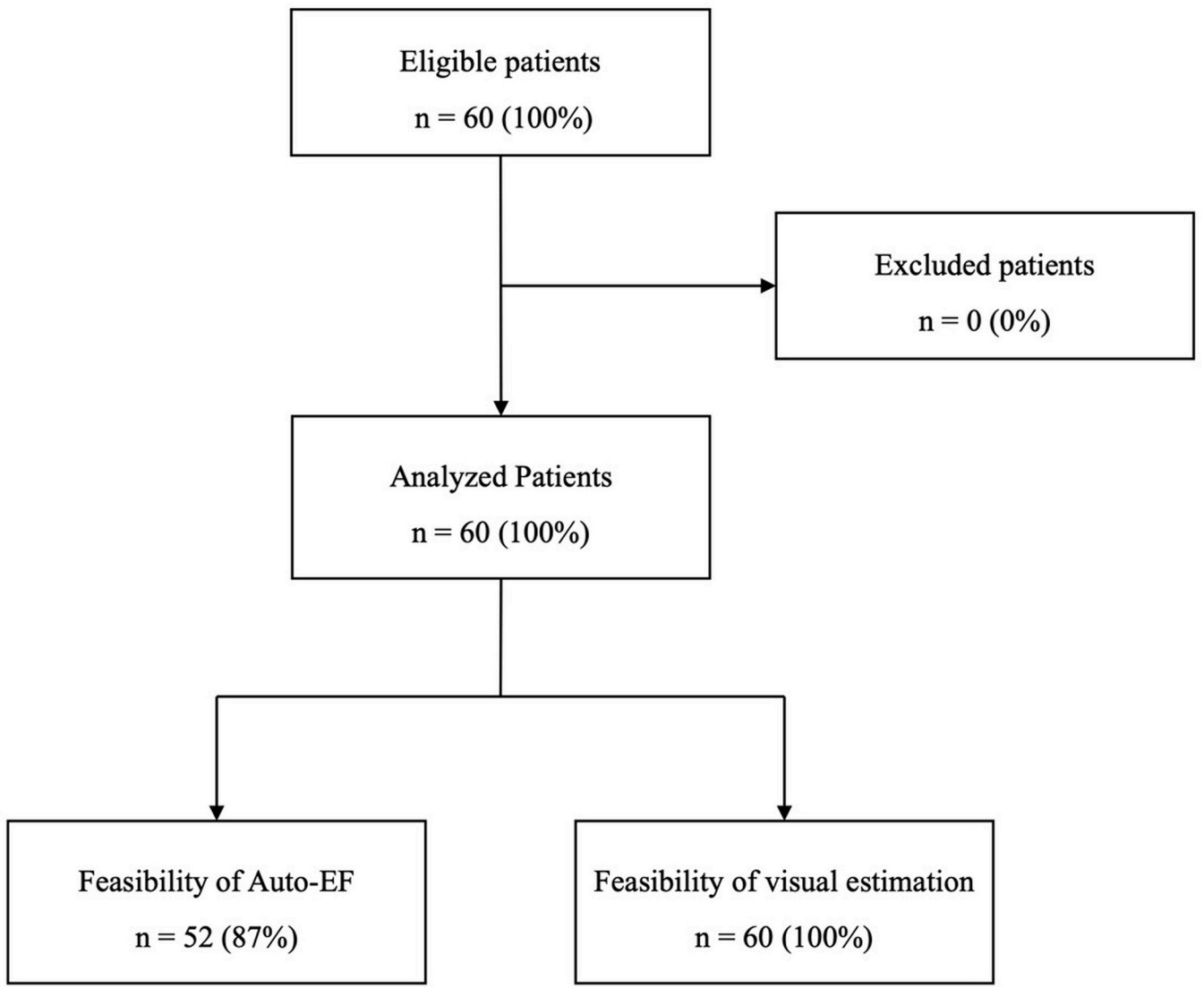

**Fig 1. Flow chart.** EF: ejection fraction.

and the LVEF estimated by cardiac MRI for classification into three stages of heart failure was 0.34 (95% CI, 0.04–0.64) (Table 2). The Pearson correlation coefficient between the peak systolic mitral annular velocity (Sa) and the LVEF estimated by cardiac MRI was 0.56 (95% CI, 0.25–0.76) (Fig 3). Concordance between Sa and the LVEF estimated by cardiac MRI for classification into three stages of heart failure was 0.15 (95% CI, −0.15–0.45) (Table 2).

## Discussion

Our study shows that concordance between auto-EF with PUD and by cardiac MRI is non-existent (Lin's coefficient = 0.23). This suggests that this tool is difficult to use in routine practice.

Although this concordance is very low, auto-EF with PUD seems capable of correctly detecting mid-range LVEF. Thus, this automatic tool, if it could be improved, could then find its place in screening for mildly impaired LVEF, where visual assessment is considered

**Table 1. Baseline characteristics of patients studied.**

| Characteristics | Mean (%) Median [Q1; Q3] |
|---|---|
| **Age** [Mean (SD)] | 62 [51; 70] |
| **Gender** (female) | 22 (37) |
| **Medical history** | |
| Chronic heart failure | 13 (22) |
| Coronary artery disease | 12 (20) |
| Diabetes-related health problems | 11 (18) |
| Hypertension | 27 (45) |
| Dyslipidemia | 23 (38) |
| BMI > 30 kg/m | 15 (25) |
| Smoking | 28 (47) |
| Valvular disease | 3 (5) |
| Heart rhythm disorders | 8 (13) |
| No medical history | 9 (15) |
| **Initial vital signs** [Mean (SD)] | |
| Systolic blood pressure (mmHg) | 133 [121; 140] |
| Diastolic blood pressure (mmHg) | 70 [61; 77] |
| Heart rate (bpm) | 70 [60; 76] |
| Respiratory rate ($min^{-1}$) | 15 [14; 16] |
| $SpO_2$ (%) | 98 [97; 100] |
| **Atrial fibrillation** | 6 (10) |
| **Indication for cardiac MRI** | |
| Diagnosis or follow-up of Cardiomyopathy | 49 (82) |
| Coronary artery disease | 11 (18) |

BMI: body mass index; $SpO_2$: oxygen saturation levels measured by pulse oximeter; MRI: magnetic resonance imaging.

imperfect. In 2019, Aldaas et al. studied the correlation between auto-EF with PUD and LVEF estimated by the Modified Simpson method in a cohort of 70 patients [15]. They found a moderate correlation (Pearson coefficient = 0.62) between the two measurements when the automatic measurement with PUD was performed by a novice sonographer with short training, and a moderate correlation (Pearson coefficient = 0.69) with an expert sonographer. Our study showed a low correlation (Pearson coefficient = 0.33) between auto-EF with PUD and the LVEF estimated by cardiac MRI. The best result in the study by Aldaas et al. needs to be qualified as the method of reference used was not the gold standard [15]. Pearson correlation coefficients were increased in a subgroup analysis that excluded poor-quality images obtained with PUD. In our study, we did not exclude any images. The improvement in correlations after exclusion of poor-quality images is explained by the difficulty for the software to delineate endocardial walls when the image is of poor quality. Ultrasound in emergency medicine is frequently performed in poor conditions and on patients in distress. This makes it difficult to obtain good-quality echocardiographic images and is a drawback for automatic assessment of EF. So, before considering routine use of the auto-EF with PUD in emergency and pre-hospital settings, its performance needs to be improved for poor-quality images. Filipiak-Strzecka et al. have also highlighted the poor performance of the auto-EF with PUD when image quality is poor [19]. Their study showed a good correlation between auto-EF with PUD and LVEF measured by software on a conventional US device, with the possibility of manually adjusting the endocardial limits when considered necessary by the expert performing the echocardiography.

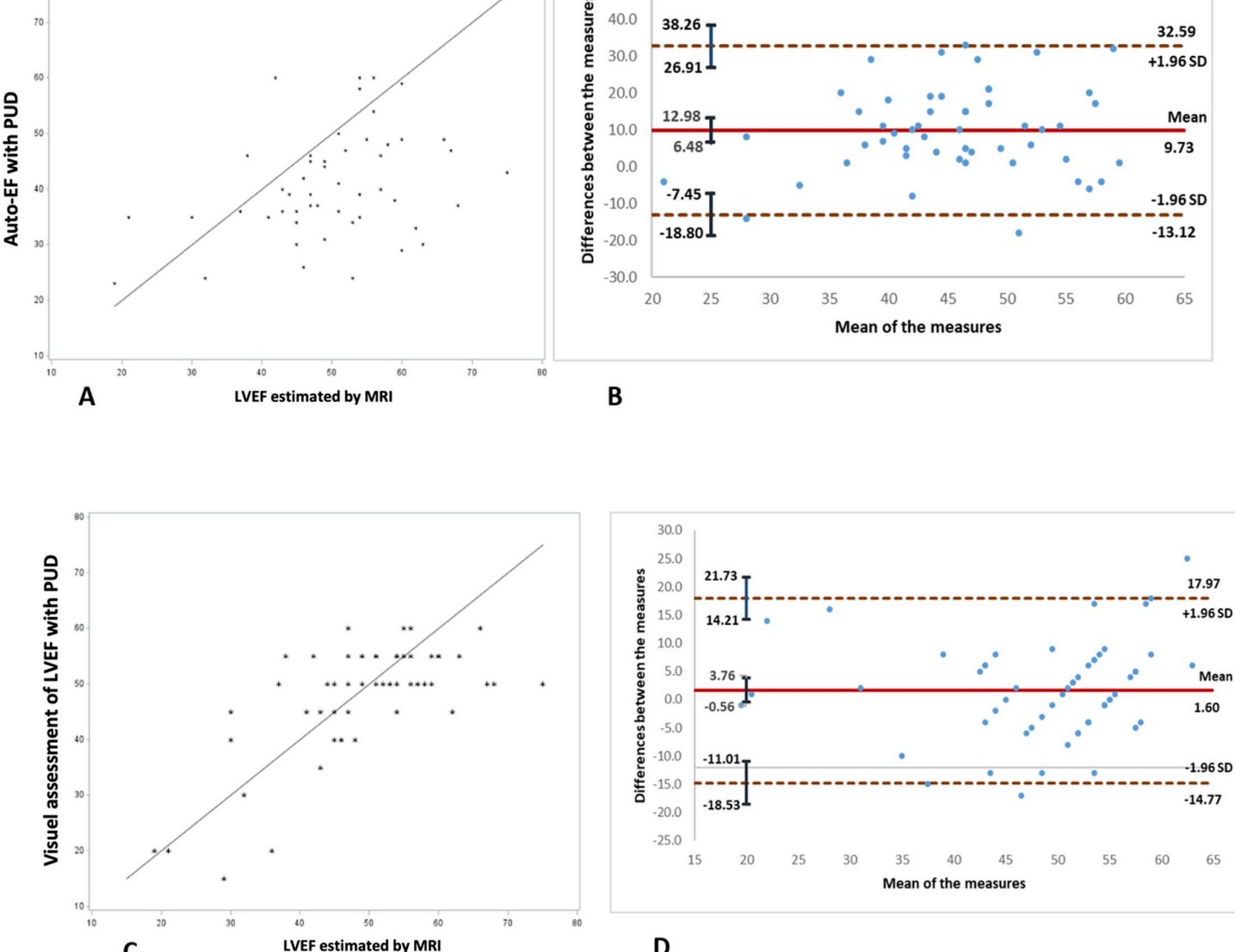

**Fig 2. Correlation between auto-EF or visual LVEF and EF estimated by MRI.** A: Correlation between Auto-EF and LVEF estimated by MRI. B: Bland & Altman plot of Auto-EF and LVEF estimated by MRI. C: Correlation between visual LVEF and LVEF estimated by MRI. D: Bland & Altman plot of visual LVEF and LVEF estimated by MRI. PUD: pocket ultrasound device; EF: ejection fraction; LVEF: left ventricular ejection fraction; MRI: magnetic resonance imaging; SD: standard deviation.

This study used echocardiography as the gold standard and not MRI, which nuances their result. Moreover, this system is based on software that operates on a single 4-chamber view and will be insensitive to regional dysfunction not seen in that view. It also requires the derivation of LV volume curves from which EF is estimated, making it quite sensitive to image quality. Furthermore, there has recently been considerable realignment in the artificial intelligence world, with new software that has its own Auto-EF based on three views, which might improve the functioning of the Auto-EF algorithm. However, adjusting the gain and flame rate are simple ways of improving image quality, and therefore the performance of automatic EF.

In 2013, Testuz et al. reported a good correlation between visual estimating LVEF with PUD and LVEF estimated by the Modified Simpson method with a conventional US device

**Table 2. Concordance between auto-EF, visual estimating, MAPSE, Sa and by cardiac MRI for classification into three stages of heart failure.**

| | | LVEF estimated by MRI (%) | | |
|---|---|---|---|---|
| | | $< 30$ | 30–49 | $\geq 50$ |
| **Auto-EF with PUD (%)** | < 30 | 1 (50%) | 2 (9%) | 2 (7%) |
| | 30–49 | 1 (50%) | 19 (86%) | 20 (72%) |
| | ≥ 50 | 0 (0%) | 1 (5%) | 6 (21%) |
| **Visual estimating LVEF (%)** | < 30 | 3 (100%) | 1 (4%) | 0 (0%) |
| | 30–49 | 0 (0%) | 12 (50%) | 2 (6%) |
| | ≥ 50 | 0 (0%) | 11 (46%) | 31 (94%) |
| **MAPSE (mm)** | < 7 | 2 (100%) | 1 (6%) | 0 (0%) |
| | 7–10 | 0 (0%) | 8 (50%) | 5 (29%) |
| | > 10 | 0 (0%) | 7 (44%) | 12 (71%) |
| Sa (cm.s$^{-1}$) | < 6 | 1 (100%) | 3 (19%) | 0 (0%) |
| | 6–8 | 0 (0%) | 7 (44%) | 7 (44%) |
| | ≥ 9 | 0 (0%) | 6 (37%) | 9 (56%) |

LVEF: left ventricular ejection fraction; MRI: magnetic resonance imaging; Auto-EF: automatic ejection fraction; PUD: pocket ultrasound device; MAPSE: mitral annular plan systolic excursion; Sa: the peak systolic mitral annular velocity.

[20]. This study agrees with our own results, which show concordance between the three heart failure classes according to EF determined by visual estimation with PUD and MRI as mean by Cohen's kappa coefficient. In our study, visual estimation detected 100% of severely impaired LVEF, 94% of normal LVEF, and 50% of mildly impaired LVEF. It therefore appears that visual estimation with PUD is a relevant tool although with a weakness in mid-range LVEF.

The originality of our study is that it used MRI as the gold standard because some of the methods used to estimate LVEF with a conventional US device correlate poorly with LVEF estimated by MRI [21, 22]. We have previously shown that visual estimation of LVEF with PUD is a very clinically relevant tool for detecting normal as well as severely impaired ejection fraction and that the automatic measurement tool could be used to detect moderately impaired ejection fraction. It would be interesting, after improving the performance of the automatic software for poor-quality images and including an option for manual adjustment of the endocardial limits, to study the concordance between this automatic tool and LVEF estimated by the gold standard (MRI), particularly for mildly reduced ejection fractions.

In our study, we observed a moderate correlation between MAPSE and LVEF estimated by cardiac MRI. This result agrees with the literature, which describes a moderate correlation between MAPSE and LVEF measured according to conventional Simpson's method [16, 23]. Like visual estimation, the original feature of our study was to compare MAPSE with LVEF measured by MRI. In addition, the lack of power in our study (35 patients), could explain why the correlation was only moderate. In addition, we did not record the images used to measure MAPSE, so we cannot rule out the hypothesis of images not conforming to the quality criteria for MAPSE measurement (alignment with the lateral mitral annulus). Finally, MAPSE could sometimes have been incorrectly measured due to kinetic disorders in patients with lateral wall dyskinesia.

Our study also shows a moderate correlation between the peak systolic mitral annular velocity (Sa) and LVEF estimated by MRI. This imperfect correlation is well established. In 2010, Park et al. found a moderate correlation between Sa and LVEF estimated with Simpson's

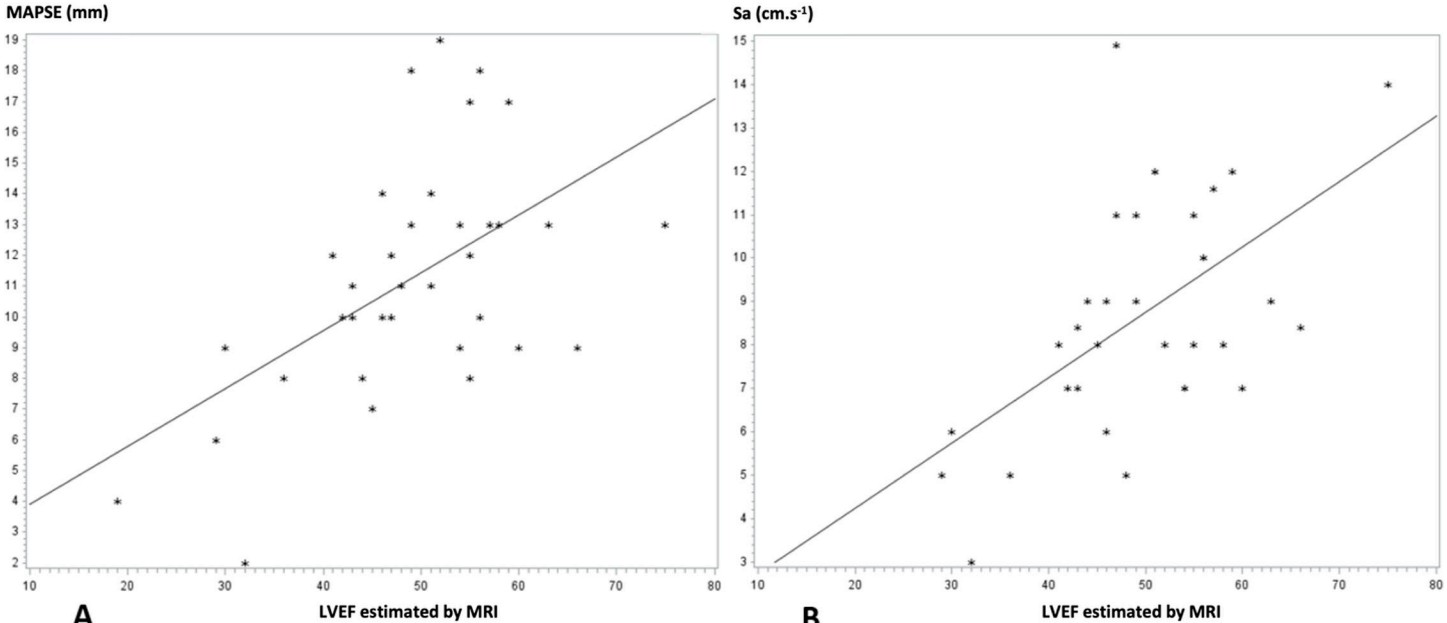

**Fig 3. Correlation between MAPSE or Sa and LVEF estimated by MRI.** A: Correlation between MAPSE and LVEF estimated by MRI. B: Correlation between Sa and LVEF estimated by MRI. MAPSE: mitral annular plane systolic excursion; Sa: peak systolic mitral annular velocity; LVEF: left ventricular ejection fraction; MRI: magnetic resonance imaging.

method [24]. In 2008, Duzenli et al. showed an absence of correlation between Sa and LVEF measured according to conventional Simpson's method in healthy patients, and a moderate correlation in patients with heart failure [9].

Our study has several limitations. First, it is a single-center study, which makes it difficult to generalize the results. Secondly, inclusions were dependent on the availability of physicians at patient admission, making this a convenience sample and therefore potentially leading to selection bias. Thirdly, our patients were outpatients and LVEF assessment with PUD is mainly used in the acute care setting. Finally, all measurements were performed by the same evaluator (auto-EF with PUD, visual estimation, MAPSE, Sa), which may introduce evaluation bias.

## Conclusion

Our study revealed non-existent concordance between the auto-EF with PUD and the LVEF estimated by cardiac MRI. The concordance between the three heart failure classes was poor. Visual LVEF concordance was mediocre and substantial by classes. These results suggest that auto-EF obtained from a single 4-chamber view using the software package used in this study was poor. It should not replace the quantitative assessment performed using an echocardiographic system and guideline-suggested algorithms to address patients' management in clinical practice. The possibility of human adjustment of the automatic delimitation of the endocardium is probably a solution for improvement.

## Supporting information

**S1 File. Minimal data set.**
(PDF)

## Author Contributions

**Conceptualization:** Lucie Berger, Xavier Bobbia.

**Data curation:** Skander Sammoud, Fabien de Oliveira, Romain Genre Grandpierre, Xavier Bobbia.

**Formal analysis:** Lucie Berger, Fabien Coisy, Fabien de Oliveira, Laura Grau-Mercier, Xavier Bobbia, Thibaut Markarian.

**Investigation:** Lucie Berger, Fabien Coisy, Skander Sammoud, Fabien de Oliveira, Romain Genre Grandpierre, Laura Grau-Mercier, Xavier Bobbia.

**Methodology:** Lucie Berger, Fabien Coisy, Fabien de Oliveira, Romain Genre Grandpierre, Laura Grau-Mercier, Xavier Bobbia.

**Project administration:** Lucie Berger, Fabien Coisy, Romain Genre Grandpierre, Xavier Bobbia.

**Resources:** Lucie Berger, Fabien Coisy, Skander Sammoud, Fabien de Oliveira, Romain Genre Grandpierre, Laura Grau-Mercier, Xavier Bobbia.

**Supervision:** Fabien Coisy, Fabien de Oliveira, Romain Genre Grandpierre, Laura Grau-Mercier, Xavier Bobbia, Thibaut Markarian.

**Validation:** Lucie Berger, Fabien Coisy, Skander Sammoud, Fabien de Oliveira, Romain Genre Grandpierre, Laura Grau-Mercier, Xavier Bobbia, Thibaut Markarian.

**Visualization:** Lucie Berger, Fabien Coisy, Skander Sammoud, Fabien de Oliveira, Romain Genre Grandpierre, Laura Grau-Mercier, Xavier Bobbia, Thibaut Markarian.

**Writing – original draft:** Lucie Berger, Xavier Bobbia, Thibaut Markarian.

**Writing – review & editing:** Fabien Coisy, Skander Sammoud, Fabien de Oliveira, Romain Genre Grandpierre, Laura Grau-Mercier, Xavier Bobbia, Thibaut Markarian.

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
