## [Decision Letter · Decision Letter 0]

12 Apr 2024

PONE-D-24-06052Evaluation of left ventricular ejection fraction by a new automatic tool on a pocket ultrasound device: concordance study with cardiac magnetic resonance imagingPLOS ONE

Dear Dr. MARKARIAN,

Thank you for submitting your manuscript to PLOS ONE. After careful consideration, we feel that it has merit but does not fully meet PLOS ONE’s publication criteria as it currently stands. Therefore, we invite you to submit a revised version of the manuscript that addresses the points raised during the review process.

We look forward to receiving your revised manuscript.

Kind regards,

Antoine Fakhry AbdelMassih

Academic Editor

PLOS ONE

Journal Requirements:

2. Thank you for submitting the above manuscript to PLOS ONE. During our internal evaluation of the manuscript, we found significant text overlap between your submission and previous work in the [introduction, conclusion, etc.].

Please revise the manuscript to rephrase the duplicated text, cite your sources, and provide details as to how the current manuscript advances on previous work. Please note that further consideration is dependent on the submission of a manuscript that addresses these concerns about the overlap in text with published work.

[If the overlap is with the authors’ own works: Moreover, upon submission, authors must confirm that the manuscript, or any related manuscript, is not currently under consideration or accepted elsewhere. If related work has been submitted to PLOS ONE or elsewhere, authors must include a copy with the submitted article. Reviewers will be asked to comment on the overlap between related submissions (http://journals.plos.org/plosone/s/submission-guidelines#loc-related-manuscripts).]

We will carefully review your manuscript upon resubmission and further consideration of the manuscript is dependent on the text overlap being addressed in full. Please ensure that your revision is thorough as failure to address the concerns to our satisfaction may result in your submission not being considered further.

"I have read the journal's policy and the authors of this manuscript have the following competing interests: TM and XB declare a competing interest as US teachers for GE (GE MEDICAL SYSTEMS ULTRASOUND) customers. The other authors state they have no competing interests."

4. In the online submission form, you indicated that [Data available on request from the corresponding author.]. 

Reviewers' comments:

Reviewer's Responses to Questions

**Comments to the Author**

1. Is the manuscript technically sound, and do the data support the conclusions?

Reviewer #1: Yes

2. Has the statistical analysis been performed appropriately and rigorously? 

Reviewer #1: Yes

3. Have the authors made all data underlying the findings in their manuscript fully available?

Reviewer #1: Yes

4. Is the manuscript presented in an intelligible fashion and written in standard English?

Reviewer #1: Yes

5. Review Comments to the Author

Reviewer #1: The author demonstrates the reliability of automatic evaluation of left ventricular ejection fraction (LVEF) on pocket ultrasound devices (PUD) by comparing with LVEF by cardiac magnetic resonance imaging (MRI). This is well-designed and well-written paper. The reviewer provides several suggestions.

<comments>

1. As the author mentioned, the PUD is useful tool in emergency medicine. However, in the field of emergency medicine, the echocardiography is often performed in bad condition. Therefore, it is often impossible to obtain good quality echocardiographic images, which is expected to be a disadvantage for automatic EF evaluation. I propose to add this viewpoint in the discussion.

2. The authors might suggest ways in which the accuracy of automatic EF could be improved, such as adjustment of gain and flame rate.

3. It is desirable to describe the method of LVEF assessment during atrial fibrillation in this study</comments>

6. PLOS authors have the option to publish the peer review history of their article (what does this mean?). If published, this will include your full peer review and any attached files.

Reviewer #1: No

---

## [Author Response · Author response to Decision Letter 0]

23 Jun 2024

Dear Editor:

We thank the Editor and the Referees for their very constructive comments and remarks. We hope that this new version might be found acceptable and addresses all the questions and concerns raised. As requested, we have attempted to answer to each referee comment and point raised. Moreover, our changes are distinguished by using a different font or color (red). Whatever your further answers, we are impressed by the attention you showed our work and the quality of your advice. We hope this new version will be worthy of your time and efforts.

Yours sincerely,

Thibaut Markarian, M.D.

 

Journal Requirements: 

An update has been provided in accordance with PLOS ONE's style requirements.

2. Thank you for submitting the above manuscript to PLOS ONE. During our internal evaluation of the manuscript, we found significant text overlap between your submission and previous work in the [introduction, conclusion, etc.]. 

Please revise the manuscript to rephrase the duplicated text, cite your sources, and provide details as to how the current manuscript advances on previous work. Please note that further consideration is dependent on the submission of a manuscript that addresses these concerns about the overlap in text with published work.

[If the overlap is with the authors’ own works: Moreover, upon submission, authors must confirm that the manuscript, or any related manuscript, is not currently under consideration or accepted elsewhere. If related work has been submitted to PLOS ONE or elsewhere, authors must include a copy with the submitted article. Reviewers will be asked to comment on the overlap between related submissions (http://journals.plos.org/plosone/s/submission-guidelines#loc-related-manuscripts).]

We will carefully review your manuscript upon resubmission and further consideration of the manuscript is dependent on the text overlap being addressed in full. Please ensure that your revision is thorough as failure to address the concerns to our satisfaction may result in your submission not being considered further. 

We apologize for any similarities in the text with previous publications. Please be assured that there is nothing intentional. We have tried our best to correct sentences that may have been inspired by other sources. However, as we don't have any specific software at our disposal, we are willing to make any necessary changes if you let us know of any similarities.

"I have read the journal's policy and the authors of this manuscript have the following competing interests: TM and XB declare a competing interest as US teachers for GE (GE MEDICAL SYSTEMS ULTRASOUND) customers. The other authors state they have no competing interests."

The declaration of competing interests has been updated in the cover letter as requested.

4. In the online submission form, you indicated that [Data available on request from the corresponding author.]. 

We take PLOS ONE's policy into consideration and have provided all data underlying the findings described as supplementary information.

We confirm that the references list is complete and correct. There are no retracted articles in this reference list.

Reviewer’s comments:

Comments to the Author 

1. Is the manuscript technically sound, and do the data support the conclusions?

Reviewer #1: Yes

Thank you for this answer.

2. Has the statistical analysis been performed appropriately and rigorously? 

Reviewer #1: Yes

Thank you for this answer.

3. Have the authors made all data underlying the findings in their manuscript fully available? 

Reviewer #1: Yes

All data underlying the findings described have been added to the supplementary information.

4. Is the manuscript presented in an intelligible fashion and written in standard English?

Reviewer #1: Yes

Thank you for this answer.

5. Review Comments to the Author 

Reviewer #1: The author demonstrates the reliability of automatic evaluation of left ventricular ejection fraction (LVEF) on pocket ultrasound devices (PUD) by comparing with LVEF by cardiac magnetic resonance imaging (MRI). This is well-designed and well-written paper. The reviewer provides several suggestions. 

Thank you for this comment. 

1. As the author mentioned, the PUD is useful tool in emergency medicine. However, in the field of emergency medicine, the echocardiography is often performed in bad condition. Therefore, it is often impossible to obtain good quality echocardiographic images, which is expected to be a disadvantage for automatic EF evaluation. I propose to add this viewpoint in the discussion. 

We agree with the reviewer's comment and have added it to the discussion.

2. The authors might suggest ways in which the accuracy of automatic EF could be improved, such as adjustment of gain and flame rate. 

We totally agree with the reviewer and have added this comment to the discussion. 

3. It is desirable to describe the method of LVEF assessment during atrial fibrillation in this study

We thank the reviewer for this pertinent comment. In this study, no adjustments were made for patients in atrial fibrillation. However, the Sa measurement corresponded to the mean of 3 measurements in the presence of atrial fibrillation. We have added this information in the “Ultrasound techniques” section.

---

## [Editor Report · Decision Letter 1]

23 Jul 2024

Evaluation of left ventricular ejection fraction by a new automatic tool on a pocket ultrasound device: concordance study with cardiac magnetic resonance imaging

PONE-D-24-06052R1

Dear Dr. MARKARIAN,

We’re pleased to inform you that your manuscript has been judged scientifically suitable for publication and will be formally accepted for publication once it meets all outstanding technical requirements.

Kind regards,

Antoine Fakhry AbdelMassih

Academic Editor

PLOS ONE
---

## [Editor Report · Acceptance letter]

1 Aug 2024

PONE-D-24-06052R1 

PLOS ONE

Dear Dr. MARKARIAN, 

I'm pleased to inform you that your manuscript has been deemed suitable for publication in PLOS ONE. Congratulations! Your manuscript is now being handed over to our production team.

Kind regards, 

on behalf of

Prof Antoine Fakhry AbdelMassih 

Academic Editor

PLOS ONE